# The Pharmacological Class Alpha 2 Agonists for Stress Control in Patients with Respiratory Failure: The Main Actor in the Different Acts

**Andreaserena Recchia** [1],*, **Maria Pia Tonti** [1], **Lucia Mirabella** [2], **Antonio Izzi** [2] and **Alfredo Del Gaudio** [1]

1   Anesthesia and Intensive Care 2, IRCCS Casa Sollievo della Sofferenza, 71013 San Giovanni Rotondo, Italy
2   Intensive Care Unit, Department of Medical and Surgical Science, University of Foggia, 71122 Foggia, Italy
*   Correspondence: a.recchia@operapadrepio.it; Tel.: +39-088-241-0703

**Abstract:** The role of sedation in patients with acute respiratory failure in the intensive care unit is crucial for improving the adaptation to mechanical ventilation, avoiding lung injury, and reducing stress related to the over-stimulated sympathetic tone. The drug class alpha 2 agonists, alone or in synergic association, can help the clinician achieve these goals. Understanding the principles of pharmacology and applying them to the alpha 2 agonists, clonidine and dexmedetomidine, can lead to different pharmaceutical choices to adapt various sedative approaches to the different stages of acute respiratory failure. A scheme is proposed using these two drugs as a pharmacological continuation for an early weaning and recovery from the intensive care unit.

**Keywords:** acute respiratory failure; alpha 2 agonists; sedation; stressors; sympathetic nervous system; synergism





## 1. Introduction

Biological stressors act on individuals to disturb normal homeostasis. Respiratory failure can occur for different causes (stressors) and elicit multiple physiological responses [1]. One response for preserving tissue oxygen supply and organ integrity is the activation of the sympathetic nervous system (SNS) as a fight or flight response. This can lead to different effects for improving the oxygen supply: increase in blood pressure and heart rate, bronchodilation, decreased intestinal activity, increased secretion of epinephrine and norepinephrine, stimulation of glucose release and gallbladder inhibition. In critical illness, protracted and over-stimulation of the SNS may cause side effects (e.g., pulmonary edema, pulmonary hypertension, heart failure, hypercoagulability, hyperglycemia, lipolysis, hyperlactatemia) [2].

The role of sedation is crucial in all patients with respiratory failure, regardless of whether they require invasive or non-invasive ventilation. Sedation is applied not only for hypnosis, muscle relaxation, and analgesia, but also to reduce the excessive activation of the sympathetic nervous system. Multiple drugs may be needed to interact with each other by enhancing desired effects and reducing side effects.

Nevertheless, according to the recent Pain, Agitation/Sedation, Delirium, Immobility and Sleep disruption (PADIS) guidelines [3], the use of strict protocols for all patients is not recommended; rather, they note the need for pain and sedation assessment and the use of lightest sedation possible, avoiding benzodiazepines and using, as a first line, propofol or dexmedetomidine. Indeed, we suggest that sedation is not isolated from other aspects of the patient. Sedation should be tailored to the different physio-pathological phases of respiratory failure, reducing stress-related patient–ventilator asynchronies and avoiding ventilator-induced lung injuries and delirium.

## 2. Alpha 2 Agonists Drugs: Principles

Alpha 2 ($\alpha$2) agonists can have a crucial role in sedation of the patient requiring ventilator support. Clonidine and dexmedetomidine are selective $\alpha$2 adrenergic receptor agonists. The sedative effect is reached through a similar mechanism in the central nervous system: presynaptic stimulation of $\alpha$2 receptors located in the prefrontal cortex and locus coeruleus, whose stimulation also reduces sympathetic tone [4]. Locus coeruleus is the primary source of release of norepinephrine in the brain and it receives and projects information from and to other brain circuits, facilitating a lot of sensory-motor and behavioral functions.

However, not all $\alpha$2 agonists lead to same response. It depends on the pharmacokinetic phase (including adsorption, distribution, metabolism, and elimination) and the pharmacodynamics phase.

### 2.1. Pharmacokinetics

Clonidine is rapidly and almost completely absorbed after oral and parenteral administration. Its oral bioavailability is 100%. It is very lipid soluble and penetrates the central nervous system. The transdermal bioavailability is 70% and it is useful during the withdrawal by continuous infusion. It is rapidly metabolized through the liver (50%) and kidney, about 60% is excreted in urine, and 20% in bile and feces [5,6]. The elimination half-life is 6–23 h and up to 41 h in patients with severe impairment of renal function. The peak action occurs in 10 min and lasts for 3–7 h after a single intravenous dose. On oral administration, it reaches a peak plasma level within 60–90 min.

According to these principles, parenteral formulations can perform the best in terms of bioavailability and dose adjustment for avoiding side effects, especially hypotension and bradycardia. The synergic administration with other sedative agents in continuous infusion can reduce the dosage and limit the side effects, thus improving sedation.

In contrast, gastric absorption can be compromised and unpredictable in the critically ill patient.

Unfortunately, the paucity of studies evaluating the efficacy of clonidine as a sedative agent on the intensive care unit precludes a systematic review [7]. Dexmedetomidine is metabolized by the liver and eliminated in urine (95%). Its elimination life is 3 h, but it is a lipophilic drug with a rapid half-life of redistribution (6 min). This explains the rapid onset and the short duration of clinical effects. It is also used for intranasal and buccal administration in pediatric premedication. High inter-individual variability in dexmedetomidine pharmacokinetics has been described, especially in the intensive care unit population. In recent years, multiple pharmacokinetic non-compartmental analyses as well as population pharmacokinetic studies have been performed. Body size, hepatic impairment, and presumably plasma albumin and cardiac output have a significant impact [8]. Results regarding other covariates remain inconclusive and warrant further research on prolonged dexmedetomidine use in continuous infusion and predictable pharmacokinetics [9]. When ECMO is required in severe ARDS, there is no evidence that increasing the dose of dexmedetomidine (>1.5 mcg/kg/h) when calculating the circuit sequestration is safer and more effective [10].

### 2.2. Pharmacodynamics Principles

Drugs can generate physiological effects depending not only on the amount of active compound that reaches receptors, but also according to the type of activated receptors and their structural changes. The pharmacological effect is reached through a dose-response relationship. The effect can be maximum (Emax) when the further increase in the concentration does not result in a higher response and it is the plateau point, moreover when plotted on a semi-log scale this relationship becomes sigmoidal (Hill-curve). Affinity, potency, and intrinsic activity are the three variables of the drug–receptor interaction [11,12]. Affinity can be defined as the extent, or fraction, to which a drug binds to receptors at any given drug concentration, or how firmly the drug binds to the receptor. Potency is a measure of the amount of the drug necessary to produce an effect of a given magnitude. The potency

of a drug is related to the EC50 (the concentration which produces 50% of the maximum effect); the higher the EC50, the lower the potency (Figure 1). Efficacy (intrinsic activity) is the ability of a drug to elicit a pharmacological response (physiological) when interaction occurs with a receptor. It is determined by the receptor interaction and by the intrinsic efficacy "$\varepsilon$", that is, the capacity of a drug to initiate a stimulus from one receptor. These three concepts (Affinity, Efficacy, and Potency) are fundamental for understanding how clonidine and dexmedetomidine drugs activate the receptor and establish concentration-response relationships with different clinical uses during the different stages of acute respiratory failure in the Intensive Care Unit.

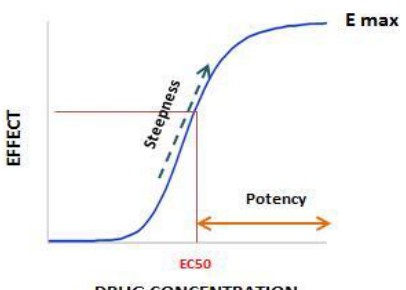

**Figure 1.** The relationship between drug concentration and effect can be explained by a sigmoidal curve. The maximum effect is reached when a further increase in concentration does not result in higher response (plateau point). The EC50 is the drug concentration required to achieve 50% of the effect and it is in relationship with drug Potency. Hill coefficient ($\gamma$) determines 'steepness' of the effect versus concentration curve, giving information on the interactions between ligand and binding sites. In fact, the curve can be steeper (if the value of $\gamma$ is greater than one) or shallower (if the value of $\gamma$ is lower than one).

### 2.2.1. The Receptors

There are three subtypes of $\alpha$2 agonist receptors: $\alpha$2 A and $\alpha$2 C are in the central nervous system (locus coeruleus and spinal cord), while $\alpha$2 B is found more in vascular smooth muscle. All the subtypes reduce adenosine monophosphate and cause hyperpolarization of noradrenergic neurons, leading to suppression of neuronal firing [13]. In the brain, receptor binding with a feedback mechanism inhibits presynaptic norepinephrine release, resulting in a reduced activity of adrenergic $\alpha$1 post-synaptic receptors in the pathways employed in arousal and in the sympathetic stress response. This mechanism leads to sedation, analgesia and a drop in blood pressure and heart rate. Spinal cord receptors are also implicated in blocking acute withdrawal symptoms in chronic opioid users. Receptors are found also in cells of the sympathetic nervous system; when they are stimulated, sympathetic nervous activity decreases with a drop in blood pressure while heart rate is often elevated in stress conditions.

### 2.2.2. Drug Interactions

Alpha 2 agonists are not always administered alone in sedation protocols. They are often associated with hypnotic and opioid drugs, resulting in interaction effects.

For example, in a sedative protocol with opioid and alpha 2 agonist drugs, the interaction in the spinal cord is synergic, resulting in greater effects than expected when the same drugs are administered alone. In synergism, less than half the dose is needed to reach the same effect, minimizing side effects, and the receptor changes occurring explain the control of acute opioid-withdrawal symptoms [14].

This can be explained because opioid and adrenergic receptors have different sites to which the drugs bind: allosteric and orthosteric sites. When the adrenergic and opioid drugs are administered simultaneously, they lead to a conformational receptor change, G-protein-mediated signal activation, an increased activation at lower opioid doses, and a change in conformation that resists downregulation. The same is found if adrenergic and

opioid drugs bind to different sites of the adrenergic receptor. Moreover, the presence of both drugs on their receptors (adrenergic drug–receptor and opioid drug–receptor) leads to the dimerization of the receptors, increasing their activation at lower doses and avoiding downregulation [15] (Figure 2).

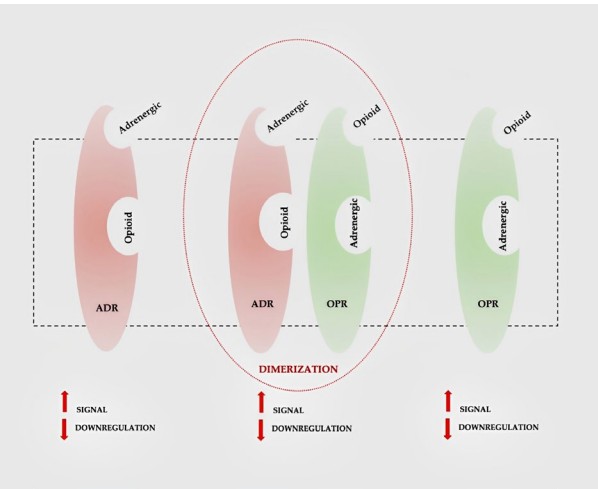

**Figure 2.** Opioid and adrenergic receptors have different sites to which both adrenergic and opioid drugs can bind. In the presence of both the ligands, not only the single receptorial conformation changes but also receptorial dimerization occurs, leading to a signal transmission increase and a receptorial downregulation reduction; this results in a synergic interaction between opioid and adrenergic drugs.

## 3. Pharmacological Scheme

According to these principles and our experience, we propose a scheme with different sedation at different stages of pathology. Sedation should be considered similarly to a theatrical work: In different acts, the main actor is always the drug class α2 agonist and its character can develop or change over the course of the story. In mild acute respiratory failure, sedation can be guaranteed by a dexmedetomidine alone infusion-based strategy; in moderate and severe acute respiratory failure, if deep sedation is required, a synergism-based strategy can occur with clonidine that guarantees the stress control and analgosedation in synergism with propofol/remifentanil. The final awake phase, with returning to spontaneous breathing, will be guaranteed by light sedation based on dexmedetomidine alone to maintain respiratory drive and delirium control (Figure 3).

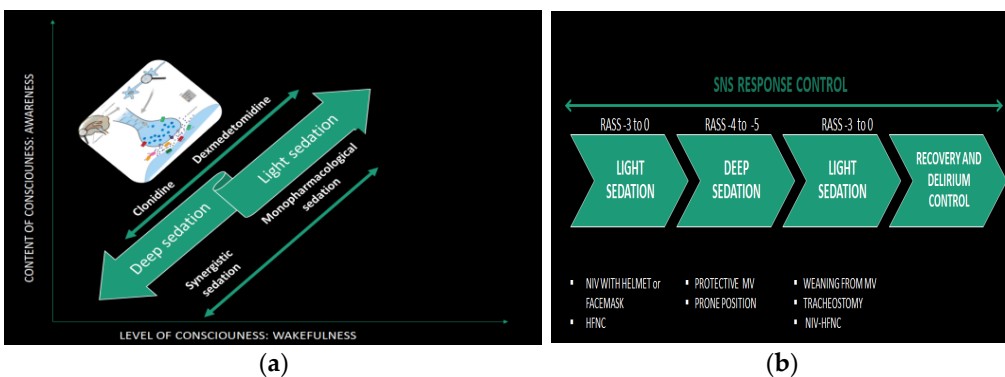

(**a**)                                                (**b**)

**Figure 3.** (**a**) α2 agonists are used throughout different stages of sedation: in deep sedation, clonidine is used with other drugs in synergic sedation; in light sedation, dexmedetomidine is used alone as a monopharmacological sedation (**b**). The aim of the scheme light sedation–deep sedation is the control of sympat hetic nervous system responses for a faster recovery from the intensive care unit with delirium control. The passage from a different sedation scale is in relation to the severity of respiratory failure and types of ventilation.

## 4. Discussion

### 4.1. Alpha 2 Agonist Drugs: Fields of Application and Side Effects

Clonidine is an alpha 2 agonist with less selectivity for alpha 2 receptors. At lower doses, it has major effects on imidazoline and α2C receptors causing hypotension, while at higher doses, it may stimulate peripheral α1 receptors causing vasoconstriction. These side effects are easily manageable and, especially at low doses, its main role is as an adjunctive and sedative sparing agent [16]. Well-designed RCTs are needed to assess the use of clonidine in ICUs [17].

DEX is an α2-adrenoceptor agonist with sedative, anxiolytic, sympatholytic, and analgesic-sparing effects, and minimal depression of respiratory function. It is potent and highly selective for α2-receptors. The alpha 2 receptor selectivity for DEX is 7-to 8-fold that of clonidine and the alpha 2: alpha 1 affinity is 1600:1. DEX is more potent and selective on alpha 2 receptors, so its role is more efficient for a light sedative effect and for sympatholytic proprieties, moving the Hill curve from right to left compared to clonidine. DEX exerts its hypnotic action through the activation of central pre- and post-synaptic α2-receptors in the locus coeruleus, thereby inducing a state of unconsciousness like restorative sleep, with slow wave sleep EEG patterns in the NREM phase [18], and the patients remain easily arousable and cooperative. DEX also has cytoprotective and anti-inflammatory proprieties [19]. Zhao et al. [20] proposed a strong rationale for using this drug to modulate inflammatory responses in COVID 19-pneumonia. DEX, by increasing parasympathetic tone and decreasing sympathetic tone, affects T cell and natural killer cells, and may suppress inflammatory response. Knowledge of the pharmacological principles should limit the side effects often associated with overdose.

In a recent Cochrane review on alpha 2 agonists for long sedation in mechanically ventilated patients, the main side effects were analyzed [7]: oversedation, bradycardia, hypotension, hypertension, tachycardia, first-degree atrioventricular block, hyperglycemia, and hypoglycemia. For clonidine, further large trials are needed, even if the main adverse effect reported in literature is bradycardia that is often dose-related and not dangerous. Bradycardia was also the most common adverse event of dexmedetomidine that emerged from the Cochrane review but, in most cases, the bradycardic effect was often mild and harmless, not requiring to be treated. Because of the small number of studies, there was insufficient power to investigate any impact of the other rare adverse events.

In a recent systematic review [21], in mechanically ventilated adults, the use of dexmedetomidine compared to other sedatives resulted in a lower risk of delirium, and a modest reduction in the duration of mechanical ventilation and ICU stay, but increased the risk of bradycardia and hypotension. Recently, Møller et al. [22] suggested the use of dexmedetomidine over other sedative agents in invasively mechanically ventilated adult ICU patients, if the desirable effects including a reduction in delirium are valued over the undesirable effects including an increase in hypotension and bradycardia.

Based on this evidence, we agree that the level of sedation, heart rate/rhythm, blood pressure, and pulse oximetry requires close monitoring during the application of sedation protocols. Clinicians should avoid a bolus and use caution, especially when these drugs are used in patients with pre-existing hypotension, hypovolemia, reduced cardiac functional reserve or impaired peripheral autonomic activity.

### 4.2. Alpha 2 Agonist Drugs in Respiratory Failure

In the last years many studies (Table 1) have been published about optimizing sedation with alpha 2 agonists in mechanically ventilated patients. Early deep sedation is often an independent negative predictor of the time to extubation, hospital death and mortality [23]. In the initial phase of acute respiratory failure, non-invasive ventilation can occur with stable hemodynamics [24]. Currently, the cornerstone of light sedation, corresponding to a RASS sedation scale from −2 to 0 (Table 2) [25], is allowing the patient to adapt himself to the non-invasive ventilation at different interfaces: High Flow Nasal Oxygen cannula (HFN) and Non-Invasive Ventilation with helmet or facemask (NIV). A better lung compliance

does the rest, if the lung injury does not evolve, and the pathology regresses thanks also to the pharmacological support. In this phase, the highly selective α2-agonist DEX may represent an optimal choice. It allows the patient to adapt to HFN or NIV while preserving optimal hemodynamic stability and maintaining the respiratory drive, reducing cardiac work and stress with the reduction of counter-regulating hormones and stress-mediated insulin resistance. Several investigations have also demonstrated the advantages of DEX in acute respiratory failure in COVID-19 pneumonia. Paternoster et al. [26], for example, used this sedative to allow awake pronation with helmet CPAP in eleven patients with COVID-19 ARDS; these were treated outside the intensive care unit with the CPAP in a prone position after failing a CPAP trial in a supine position, and showed evidence of an increase of Spo2, Pao2/Fio2 ratio and a reduction of respiratory rate. Nevertheless, when a further increase of the lung damage with moderate or severe ARDS or hemodynamic instability occurs, invasive mechanical ventilation becomes necessary, and the light sedation scheme can fail, requiring deep sedation (RASS sedation scale −3 to −5) for improving pulmonary compliance and suppressing ventilatory drive. This degree of sedation is also required to avoid unintentional self-extubation and to guarantee the prone position when required.

**Table 1.** Summary of some studies on sedation in mechanically ventilated patients.

| Author, Year | Study Type | *n* | Drug | Results |
|---|---|---|---|---|
| Chen, 2015 [27] | Systematic review | 7 trials, 1624 pts | Dexmedetomidine/clonidine | Dexmedetomidine reduces duration MV and ICU length of stay. No evidence in reducing delirium and overall death rate. Slow heartbeat incidence is doubled. More studies are needed for clonidine |
| Cruickshank, 2016 [17] | Systematic review | 18 trials, 2489 pts | Clonidine/dexmedetomidine/propofol/benzodiazepines | Evidence on the use of clonidine is very limited. Dexmedetomidine may be effective in reducing ICU length of stay and time to extubation. Risk of bradycardia, but not overall mortality, is higher among patients treated with dexmedetomidine. |
| Wang, 2017 [16] | Systematic review and metanalysis | 8 trials, 642 pts | Clonidine/other sedatives | Clonidine is a narcotic sparing agent but increases incidence of hypotension. Data on clonidine are insufficient. |
| Shehabi, 2019 [28] | Randomized controlled trial | 4000 pts | Dexmedetomidine/usual standard sedatives | Rate of death in dexmedetomidine is similar to standard care group, but more sedatives are required and more adverse effects occur. |
| Shehabi, 2021 [29] | Bayesan analysis of clinical trial | 3904 pts | Dexmedetomidine/usual standard sedatives | Dexmedetomidine can reduce mortality in older patients and increase it in younger patients (≤65 years) with non post-operative status. |
| Page, 2021 [23] | Narrative review | Sedation literature over the 5 previous years | Dexmedetomidine/clonidine/other drugs | Except for dexmedetomidine, more attention in literature is required for other drugs. |
| Lewis, 2022 [21] | Systematic Review | 77 trials, pts | Dexmedetomidine/GABAergic drugs | Dexmedetomidine use results in a lower risk of delirium and a modest reduction in duration of MV and ICU stay, but the risks of bradycardia and hypotension are increased |

Abbreviations: MV mechanical ventilation; ICU intensive care unit.

**Table 2.** Richmond Agitation-Sedation Scale (RASS). The Richmond Agitation and Sedation Scale is a validated and reliable method to assess patient's level of sedation in the intensive care unit.

| Score | Term | Description |
| --- | --- | --- |
| +4 | Combative | Overtly combative, violent, immediate danger to staff |
| +3 | Very agitated | Pulls or removes tube(s) or catheter(s); aggressive |
| +2 | Agitated | Frequent non-purposeful movement, fights ventilator |
| +1 | Restless | Anxious but movements not aggressive or vigorous |
| 0 | Alert and Calm | |
| −1 | Drowsy | Not fully alert, but has sustained awakening (eye-opening/eye contact) to voice (>10 s) |
| −2 | Light sedation | Briefly awakens with eye contact to voice (<10 s) |
| −3 | Moderate sedation | Movement or eye opening to voice (but no eye contact) |
| −4 | Deep sedation | No response to voice, but movement or eye opening to physical stimulation |
| −5 | Unarousable | No response to voice or physical stimulation |

The recent SPICE III study [28] was a randomized clinical trial comparing the effect of early light sedation with dexmedetomidine on all-cause mortality with the effect of "usual standard of care" in 4000 ventilated and intubated critically ill adult intensive care unit (ICU) patients. The sedation goal was RASS from −2 to +1.

The dexmedetomidine group had a rate of death at 90 days similar to that in the usual care group, but required more sedatives to achieve the prescribed level of sedation and experienced more adverse effects. Unfortunately, more than 60% of patients had an indication to deep sedation and 11.5% of the usual group received dexmedetomidine for achieving light sedation despite the protocol. In a secondary analysis of the SPICE III trial [29], dexmedetomidine was associated with an increased risk of mortality in the age group ≤65 years compared with alternative sedatives. This heterogenous effect on mortality from age was most prominent in patients admitted for reasons other than post-operative care, and increased with increasing APACHE II scores and with decreasing age. The authors concluded that the mechanism is not known, and that further studies are needed. From the SPICE III trial, it emerged that when dexmedetomine is added to midazolam or propofol to reach deep sedation, it increases adverse events. Clinicians should use caution in giving dexmedetomidine during sedation for patients requiring deep or moderate sedation in mechanical ventilation, especially in younger patient (≤65 years) with non-post operative respiratory failure and higher (≥25) APACHE 2 scores. Instead, the preference would be to use the poly-pharmacological scheme with clonidine, hypnotics and opioids in synergic association, limiting dexmedetomidine use alone in non-invasive ventilation and weaning from mechanical ventilation.

Consequently, clinicians propose that a multimodal patient-centered approach is required, but the combination and dosage of drugs used must be appropriately calibrated [30].

According to the SPICE III trial results, we do not propose dexmedetomidine for poly-pharmacological association because its receptorial affinity, potency and selectivity can increase adverse effects, especially hemodynamic ones. According to the literature, we remark an early goal direct sedation and the use of dexmedetominine during light sedation in mechanical and intubated patients just if used alone especially in less severe illness and il elderly when light sedation can occur. When a deeper sedation is necessary, clonidine is the right drug. It has fewer sedative properties and can reduce the hypnotic dose of other drugs, guaranteeing improved hemodynamic stability, less burst suppression, withdrawal symptoms and delirium prevention [31]. Thus, the cornerstone in treatment becomes drugs synergism [32]. Subsequently, as respiratory mechanics improve, it is necessary to reduce the level of sedation [33]. In this state, a return to the strategy of light sedation can reduce

the time to extubation with beneficial effects. Since sedatives with drug accumulation effects should be avoided, DEX alone appears to be the best choice. The effect is a shorter duration of MV and improved patient comfort, with a better control not only of heart rate, blood pressure, ventilator synchronism, metabolic and anti-inflammatory function, but also of delirium, which is a neurobehavioral syndrome caused by dysregulation of baseline neuronal activity secondary to systemic disturbances. This strategy can be used also when tracheostomy becomes necessary based on a difficult weaning [34].

## 5. Conclusions

Choosing and optimizing sedation aimed at patient synchronism with mechanical ventilation is crucial to avoid not only ventilator-induced lung injuries but also to reduce stress-related patient–ventilator asynchronies. The α2 agonist drugs can be used alone or in synergistic association, but adequate knowledge of the pharmacokinetic and pharmacodynamics principles is mandatory for the clinician. These drugs not only have a sedative effect, but also other properties due to the interaction with the SNS that can be overstimulated when normal homeostasis is disturbed by critical illness. We agree with considerations of authors Constantin et al. [35]: for reaching an objective, the sedation protocol should be inserted in a complex intervention, including assessment of pain and sedation, spontaneous awakening and respiratory trials, delirium control, early mobilization, family engagement and, we add, stress control.

The strategy of sedation with α2 agonist drugs, tailored to the different physio-pathological phases of respiratory failure, not only considers the hypnotic and analgesic effects, but also the methods of administration and the synergic pharmacological interactions, aiming at the control of the homeostasis for an adequate recovery from the intensive care unit.

**Author Contributions:** Conceptualization, A.R. and A.D.G.; resources, M.P.T. and A.I.; supervision, L.M. and A.D.G.; writing—original draft, A.R.; Writing—review & editing, A.R. All authors have read and agreed to the published version of the manuscript.

**Funding:** This research received no external funding.

**Institutional Review Board Statement:** Not applicable.

**Informed Consent Statement:** Not applicable.

**Data Availability Statement:** No new data were created or analyzed in this study. Data sharing is not applicable to this article.

**Conflicts of Interest:** The authors declare no conflict of interest.

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
