# Peer review of "The Pharmacological Class Alpha 2 Agonists for Stress Control in Patients with Respiratory Failure: The Main Actor in the Different Acts"

_stresses, doi:10.3390/stresses3010001_

Round 1

Reviewer 1 Report

This manuscript describes an interesting theme in sedation for patients under respiratory failure. However, it does not advance our knowledge any more. There is no merit for publication. 

Reviewer 2 Report

Thank you fort he opportunity to review the narrative review „The pharmacological class alpha 2 agonists for stress control in patients with respiratory failure: the main actor in the different acts“. Alpha 2 agonists are an important class of pharmacological substances used in critically ill patients. 

Comments and suggestions:

Elaboration on side effects of Alpha 2 agonists is needed. Especially hemodynamic side effects need to be addressed e.g. potentially significant clonidine-induced bradycardia and/or hypotension.

Are there contraindications to use these drugs?  

Lately, there are data showing an increased dexmedetomidin-associated mortality risk in ICU patients of an age 65 years (SPICE III study). Therefore, in Europe official organs recommend against the use of dexmedetomidin in ICU patients 65 years of age. This is a severe limitation of the proposed pharmacological sedation scheme.

In the light of the recent evidence on dexmedetomidin in younger ICU patients – are the recommendations of the authors regarding sedation management still justified by the paradigms of current standard of care?

Would the authors propose a patient-individual sedation concept? Different sedation concepts for patients > and 65 years?

In patients with respiratory failure, a wide range of invasive and non-invasive ventilation modes exists. Are there different roles for clonidine in modes that allow spontaneous breathing during controlled ventilation in severe ARDS.

What is the role of Clonidine and dexmedetomidin in patients treated with ECMO? 

Is the use of alpha 2 agonists in spontaneously breathing ICU patients with mild ARDS and dyspnoe superior compared to the use of morphine?

Please revise complete manuscript for English language and grammar. I strongly recommend the authors work with a native English speaker to work through quite a few poorly worded and confusing sentences.

In addition, inappropriate wording is used multiple times.

Deep sedation rather than heavy sedation and without abbreviation

223-226: reference missing

Reviewer 3 Report

Dear authors,

Thank you for submitting your work. Please go through my comments and send a revised manuscript at the earliest.

Arrange keywords in alphabetical order.

3. Pharmacological scheme- The authors have mentioned about their experience. The readers would want to know their experience in the form of some outcome data, even if not published already.

The review of literature is not comprehensive in the discussion (Ex: Several investigations demonstrate the advantages of DEX also in acute respiratory failure in COVID-19 pneumonia. 215 Paternoster et al. [19], for example, used this sedative to allow awake pronation with hel-216 met CPAP outside the ICU.) The authors should mention some corroborative details of the study to make a point.

In a review article, a table summarizing important studies makes a good read and also enhances the quality of the article.

It is better to mention the profile of oral clonidine along with IV clonidine and to explain if its comparable, and also cost-effectiveness, effect on hemodynamics.

Round 2

Reviewer 1 Report

The manuscript was revised and some new information was added. However, information provided in the manuscript remains less useful in a clinical setting for the stress treatment of patients with respiratory failure.

Reviewer 3 Report

Dear authors,

Thank you for making changes in the manuscript based on my suggestions. I can see substantial change in the present write-up when compared to previous one. Well done.